# Long-Read Metagenomics Improves the Recovery of Viral Diversity from Complex Natural Marine Samples

Asier Zaragoza-Solas,[a] Jose M. Haro-Moreno,[a] Francisco Rodriguez-Valera,[a] Mario López-Pérez[a]

[a]Evolutionary Genomics Group, División de Microbiología, Universidad Miguel Hernández, San Juan, Alicante, Spain

**ABSTRACT** The recovery of DNA from viromes is a major obstacle in the use of long-read sequencing to study their genomes. For this reason, the use of cellular metagenomes ($>0.2$-$\mu$m size range) emerges as an interesting complementary tool, since they contain large amounts of naturally amplified viral genomes from prelytic replication. We have applied second-generation (Illumina NextSeq; short reads) and third-generation (PacBio Sequel II; long reads) sequencing to compare the diversity and features of the viral community in a marine sample obtained from offshore waters of the western Mediterranean. We found that a major wedge of the expected marine viral diversity was directly recovered by the raw PacBio circular consensus sequencing (CCS) reads. More than 30,000 sequences were detected only in this data set, with no homologues in the long- and short-read assembly, and ca. 26,000 had no homologues in the large data set of the Global Ocean Virome 2 (GOV2), highlighting the information gap created by the assembly bias. At the level of complete viral genomes, the performance was similar in both approaches. However, the hybrid long- and short-read assembly provided the longest average length of the sequences and improved the host assignment. Although no novel major clades of viruses were found, there was an increase in the intraclade genomic diversity recovered by long reads that produced an enriched assessment of the real diversity and allowed the discovery of novel genes with biotechnological potential (e.g., endolysin genes).

**IMPORTANCE** We explored the vast genetic diversity of environmental viruses by using a combination of cellular metagenome (as opposed to virome) sequencing using high-fidelity long-read sequences (in this case, PacBio CCS). This approach resulted in the recovery of a representative sample of the viral population, and it performed better (more phage contigs, larger average contig size) than Illumina sequencing applied to the same sample. By this approach, the many biases of assembly are avoided, as the CCS reads recovers (typically around 5 kb) complete genes and even operons, resulting in a better discovery of the viral gene diversity based on viral marker proteins. Thus, biotechnologically promising genes, such as endolysin genes, can be very efficiently searched with this approach. In addition, hybrid assembly produces more complete and longer contigs, which is particularly important for studying little-known viral groups such as the nucleocytoplasmic large DNA viruses (NCLDV).

**KEYWORDS** PacBio CCS long reads, bacteriophage, long-read sequencing, metagenome, viral diversity, virome

Address correspondence to Mario López-Pérez, mario.lopezp@umh.es, or Francisco Rodriguez-Valera, frvalera@umh.es.

The authors declare no conflict of interest.

**M**arine viruses are the most abundant biological entities in oceanic marine environments, with an estimated population density of $10^7$ per mL of seawater (1). It is therefore no wonder that they are critical drivers of ocean biogeochemistry, both via the release of organic matter as a by-product of their predation upon phytoplankton and heterotrophic bacteria (2, 3) (viral shunt) and via the manipulation of host metabolism during infection (4, 5).

During the last 2 decades, the study of the viral community in marine environments has been driven by metagenomics, thanks to the advances in short-read (SR) sequencing (6, 7). However, the advent of long-read (LR) sequencing technology, spearheaded by Oxford Nanopore and Pacific Biosciences (PacBio), has the potential to solve major issues that have plagued SR sequencing-based studies for years, mainly the low recovery of both high-diversity microbes (8) and the flexible genome (9). Unfortunately, the high error rate derived from these technologies has delayed their application in metagenomics and requires the use of either a complementary short-read data set (10) or very high coverage (11) to correct these sequencing errors. The development of high-fidelity approaches such as PacBio circular consensus sequencing (CCS) with an error rate similar to that of the Illumina system opens a new avenue for the study of prokaryotic communities in their natural environments. An example of the advantages of this new technology can be found in a recent work, in which a well-known marine sample from the Mediterranean water column was analyzed using both SR and LR sequencing (12). Results suggested that PacBio Sequel II CCS is particularly suitable for cellular metagenomics due to its large read size and its low error rate. Reads in LR metagenomes are large enough to perform gene prediction directly, bypassing the biases inherent in the assembly process. The assembly step is also improved with this kind of sample by using hybrid assembly of LR and SR, allowing reconstruction of genomes, including the flexible genome and even streamlined genomes, such as those from *Pelagibacterales* (12).

The benefits of LR sequencing can be even more pronounced for the study of viruses, as the size of individual reads may be sufficient to recover complete genomes. There are already some examples of LR sequencing applied to the study of viromes using the Nanopore sequencing platform. Beaulaurier et al. recovered 1,864 new complete assembly-free virus genomes from three Nanopore data sets (11). On the other hand, Warwick-Dugdale et al. recovered around 2,500 viral contigs from the assembly of Nanopore and Illumina data sets from the same seawater sample of the western English Channel, showing that a hybrid or long-read-only assembly improved the recovery of viral contigs and their metaviromic islands compared to short-read assemblies (10). These results have been corroborated in other virome studies using the same technology (13, 14).

However, the study of viromes by LR sequencing is limited by the large amount of DNA required for this type of technology and the scarcity of viral DNA that can be collected from environmental samples. Therefore, as an alternative to the study of the virome, we used the viral DNA present in a cellular metagenome ($>$0.22-$\mu$m size range). A high presence of viral DNA (around 10% to 15%) in marine metagenomes has been reported (15). The vast majority of this viral DNA likely belongs to cells undergoing the lytic cycle, although other sources might be possible, including lysogenized viruses (either integrated or as a plasmid) or virions larger than the filter pore ($>$0.2 $\mu$m) (15). The aim of this study was to compare the efficiency of LR sequencing for the study of the viral community (with and without an assembly step) with the classical approach using Illumina (short reads).

## RESULTS AND DISCUSSION

To evaluate the viral genomic diversity resolution power of LR metagenomics and compare it to that of SR sequencing, we analyzed a single marine sample from offshore Mediterranean waters during winter, when the epipelagic water column was mixed. The presence of replicating viruses inside cells during the lytic cycle produces a natural amplification that makes it possible to find abundant sequences of viral origin in the cell fraction of metagenomic samples. This sample was sequenced with Illumina and PacBio Sequel II systems and then assembled twice, first using only the Illumina short reads, resulting in the short-read assembly data set (SRa), and then in a hybrid assembly using both the Illumina short reads and the PacBio long reads, resulting in the long-read assembly data set (LRa). We decided on the hybrid assembly rather than a long-read-only assembly based on previous results (12). In order to evaluate the possible biases introduced by the assembly process, we also analyzed the PacBio CCS15

**TABLE 1** Summary statistics of viral sequence recovery for the short-read assembly (SRa), long-read assembly (LRa), and raw-read (LR) data sets

| Statistic | Illumina assembly (SRa) | PacBio assembly (LRa) | PacBio CCS15 reads (LR) |
|---|---|---|---|
| Starting sequences | 149,018 | 19,982 | 1,535,891 |
| Putative phages (VIBRANT) | 10,979 | 947 | 50,296 |
| 95% identity clustering | 10,979 | 947 | 42,156 |
| Unique sequences[a] | 5,886 | 36 | 30,203 |
| Nucleotides sequenced (Gb) | 23.4 | 31.0 | 7.6 |
| Unique sequences/Gbp sequenced | 251.53 | 1.16 | 3,974 |
| Unique sequences (versus GOV2)[b] | 4,196 | 35 | 26,766 |
| No. complete (high quality)[c] | 9 (53) | 15 (114) | 0 (27) |
| Min–max sequence length (bp) | 1,000–188,349 | 1,353–428,169 | 1,011–17,836 |
| Avg sequence length (bp) | 4,906 | 32,260 | 5,261 |
| Min–max GC content (%) | 19.40–65.25 | 19.56–69.93 | 14.25–86.03 |
| Avg GC content (%) | 35.45 | 36.9 | 38.13 |
| Total proteins[d] | 80,487 | 41,599 | 330,157 |
| Unique terminase (terL) proteins | 30 | 2 | 393 |
| Avg proteins/sequence | 7.33 | 43.92 | 7.83 |
| Avg protein length (aa) | 190.29 | 223.42 | 177.9 |

[a]Sequences not present in the other data sets (BLASTN, 95%; coverage of at least 70% of the smallest sequence).
[b]Sequences not present in the other data sets or the Global Ocean Virome 2.0 (BLASTN, 95%; coverage of at least 70% of the smallest sequence).
[c]VIBRANT defines a high-quality sequence as one that likely contains the majority of a virus's complete genome (~70% completeness).
[d]Values shown here represent protein numbers after dereplication (CD-HIT, 95% identity).

reads (PacBio consensus reads created by comparing at least 15 subreads [LR]) before assembly.

**Viral sequence recovery and statistics.** First, we wanted to compare the efficiency of viral sequence recovery between the three data sets (Table 1). The first step in the preprocessing pipeline was to run VIBRANT (16) for all sequences >1 kb to identify those in each data set that were of viral origin. Viral sequences turned out to be quite numerous in both data sets, with 5% of the total sequences from the SRa and LRa and 2.5% of the LR data set classified as viral contigs. After a step of clustering at 95% sequence identity to remove redundant reads from the LR data set, we recovered a total of 54,082 putative viral sequences (10,979 in the SRa, 947 in the LRa, and 42,156 in the LR) (Table 1). In order to assess if the different assembly methods recovered the same viral community, we identified unique sequences in each data set by comparing the three data sets against each other (see Materials and Methods). Most sequences from the LRa were also found in the SRa, with only 36 unique LRa contigs. Remarkably, while the SRa data set contained a fair number of unique sequences (5,886), most of the unique sequences were found in the LR data set (30,203; 71% of total viral LR sequences), revealing a large genomic diversity not recovered by the assemblies. This diversity gap was also present when results for a marker gene, such as that encoding the terminase large subunit (terL), were compared, with the LR data set containing 393 unique terminase genes (clustering at 95% amino acid identity), compared to 30 and 2 in the SRa and LRa data sets, respectively. The GC content showed a slight (effect size = 0.022) but significant (Kruskal-Wallis test, $P$ value $< 10^{-15}$) skew toward high GC values when PacBio CCS reads were added to the data sets (Table 1). The SRa data set presented an average GC content of 35.45% compared to 36.9% for the LRa and 38.13% for the LR (Table 1). This bias could arise from the fact that assemblies usually recover only the core genome. In this sample (marine surface water), clade SAR11 is the most abundant organism (12), with an average GC content of 34%. LRs recover more of the flexible genome, which can present GC fluctuations compared to the core and would thus explain this variation from 34% to 38%. Regarding sequences shared between the three data sets, Table S1 shows the relationship between contigs that were considered part of the same phage (identity over 95%, 70% overlap of the

smallest contig). When comparing the ratio of recovered sequences between SRa and the combined LRa and LR data sets for shared ones, we found that in 2,463 of 3,316 shared instances (ca. 75%), the LR data sets contained longer contigs than their SRa counterpart (Table S1). These results show that the use of long reads in assembly result in larger contigs than assembly with only SR.

Next, we were interested in assessing if this novel diversity had been captured by previous studies, so we compared the three data sets against the Global Ocean Virome 2 (GOV2) (17), the largest database of seawater phages to date (195,728 marine populations, containing 6,685,706 proteins). This data set was created from viromes obtained from 145 samples from the Malaspina (18), *Tara* Oceans (6) and *Tara* Arctic (17) expeditions, therefore representing marine phage communities from different environments all around the world. We found 30,997 viral sequences in our whole data set (SRa, LR, and LRa) not found in GOV2, with the vast majority (26,766) of these unique sequences belonging to the LR data set.

Regarding size and completeness, the hybrid PacBio LRa resulted in the largest viral contigs, with a maximum size of 428,169 bp and an average contig size of 32,260 bp (Table 1). We recovered 24 complete phage genomes (based on circular redundancy at the ends) from both assembled data sets (15 in the LRa and 9 in the SRa). As expected, due to their small estimated average size (ca. 5 kb), we were unable to recover any complete genomes directly from the LRs. However, we can make an estimated guess of the quality of the remaining contigs using VIBRANT's quality statistics, which classify contigs based on the estimated completeness of the genome. When we considered only contigs marked as high quality (70% of the estimated phage genome), we found that only 53 (0.4%) of the SRa contigs belonged to this category, while in the LRa data set there were 114 (12.5%) (Table 1). Some complete phage genomes were shared by the LRa and SRa data sets. The SRa contigs resulted in a maximum contig size approximately half of that found in the LRa (188,349 bp), with an average contig size on a par with the LR data set, more than six times smaller than the average in the LRa (ca. 32 kb) (Table 1). These results, together with the facts that the average protein size in all three data sets is similar and the number of proteins recovered from the LR data set is an order of magnitude larger than those from the assembled data sets, suggest that PacBio CCS15 reads could be used for viral protein calling without the need for assembly, as previously stated (12).

**Putative host prediction.** An important part of the biological significance of viruses depends on knowledge of the host they infect. We attempted to assign a host to contigs in all three data sets (SRa, LRa, and LR). To this end, phage contigs were classified against the RefSeq database. We assigned hosts to each sequence following the method described by Beaulaurier et al. (11), which was applied to phages obtained by Nanopore sequencing. The method is based on protein homology against a reference database, assigning a host to a sequence based on the number of best hits (see Materials and Methods). Figure 1A shows the results at a 3-protein threshold, including all contigs from a data set and those unique to their specific data set (dereplicated) and before dereplication. Considering the SRa and LR data sets, both unique and dereplicated variants presented a similar host assignment rate (ca. 30%) with no differences at the taxonomic level, suggesting that the differences between the two data sets could be beyond the order level. As might be expected, *Alphaproteobacteria* and *Cyanobacteria* were the most abundant hosts in all three data sets, as they were the most abundant groups in the sample (12) and were also the most represented in the reference databases (Fig. 1A). The recent addition of various *Methylophilales* (19) and *Flavobacteria* (20) phage genomes to the reference databases has resulted in a highly increased *Gammaproteobacteria* and *Flavobacteria* phage count compared to previous analysis of the Mediterranean virome (7).

The LRa data set provided the highest rate of host assignment. In the nondereplicated sample, almost 75% of the contigs had a host assigned, compared to a 30% rate for the LR and SRa data sets. This is probably due to the fact that they were, on

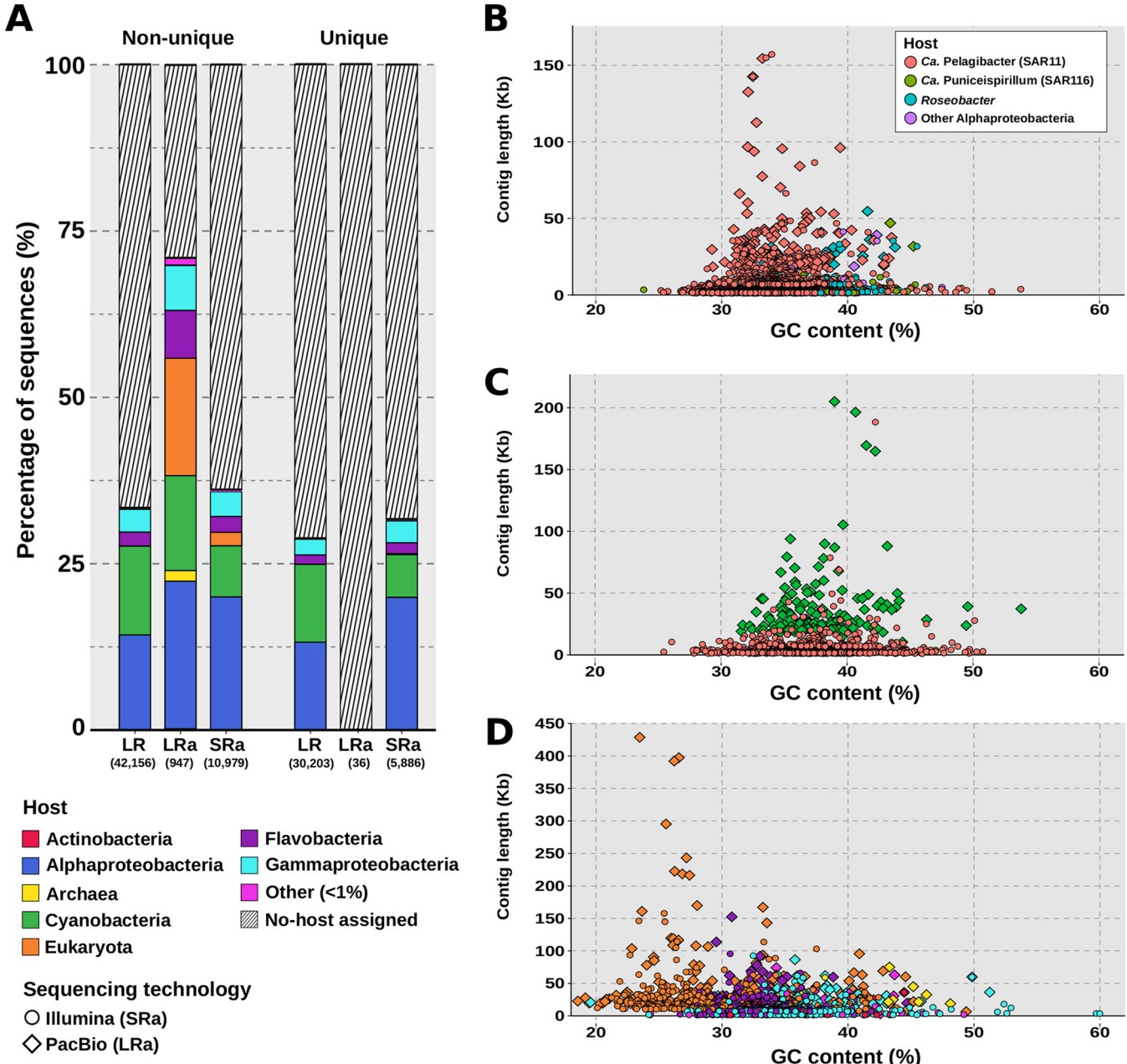

**FIG 1** (A) Taxonomic affiliations of viral contigs expressed in percentages, separated into those found in that data set (non-unique) and those unique to that data set (unique). The number in parentheses below each bar is the number of contigs in that category. (B) Distribution of assembled viral contigs that infect *Alphaproteobacteria* by contig length and GC content. Circles represent short-read assemblies (Illumina), while diamonds represent hybrid assemblies (PacBio + Illumina). Shapes are colored according to their host. (C) Distribution of assembled viral contigs that infect *Cyanobacteria* by contig length and GC content. Orange circles represent short-read assemblies (Illumina), while green diamonds represent hybrid assemblies (PacBio + Illumina). (D) Distribution of viral contigs by contig length and GC content. Circles represent short-read assemblies (Illumina), while green diamonds represent hybrid assemblies (PacBio + Illumina). Shapes are colored according to their host.

average, larger contigs and as such contain more information to reliably assign a host. However, we were unable to assign a host to any of the 36 unique sequences in the LRa data set (3.2% of the total). Host taxonomy was similar to that seen in the previous data sets, the main difference being an increase in eukaryotic and archaeal viruses (20% of total contigs), mainly marine group I *Thaumarchaeota* (marthavirus) (21).

Comparison between the sequences obtained by assembly (LRa and SRa) also revealed differences between the viral groups. As a general rule, LRa contigs were on

average larger than their SRa counterparts, even if the latter can result in similar maximum sizes. For example, in alphaproteobacterial phages (Fig. 1B), we recovered 52 sequences over 30 kb in the SRa data set, compared to 126 in the LRa data set. We found a similar case for the cyanophages (Fig. 1C), where 14 sequences were over 50 kb in the SRa data set compared to 68 sequences in the LRa data set. The nucleocytoplasmic large DNA viruses (NCLDV, proposed order *Megavirales*) (Fig. 1D) deserve special attention, as their assemblies in the LRa data set were larger and more numerous (24 sequences over 20 kb, including the largest contig of 428 kb) than those in the SRa data set (21 sequences, 2 over 50 kb; maximum size, 61 kb). We believe this might be due to the fact that eukaryotic genomes have many repeats and other features that make their assembly from short-read metagenomes less efficient (22).

To analyze the phylogenomic diversity of the NCLDV sequences found, we used only sequences that contained five key markers highly conserved in this type of virus: the major capsid protein (MCP), the DNA polymerase beta subunit (PolB), the DEAD/SNF2-like helicase SFII, the poxvirus late transcription factor VLTF3, and the packaging ATPase A32 (23). Figure S1 shows a phylogenetic tree based on a concatenation of these five proteins, including reference genomes from RefSeq and the collection of 444 marine NCLDV Metagenome-Assembled Genomes from the work of Moniruzzaman et al. (23). The tree shows that these new eukaryotic sequences fall in the family *Mimiviridae* (16 sequences) and the family *Phycodnaviridae* (8 sequences).

**Relative abundance in marine samples.** Next, we wanted to analyze whether all the diversity found only in the LR data set was abundant and representative in nature. For that reason, we performed a recruitment analysis of SRa, LRa, and dereplicated LR viral sequences against the entire *Tara* Oceans metagenome data set (24). We considered a sequence present in a metagenomic sample if the sequence recruited at least five reads per kilobase of sequence and gigabase of metagenome (RPKG), with an identity of 95% and a contig coverage of 50%. The results are shown in Fig. 2. Although pelagiphages and cyanophages (viruses that infect "*Candidatus* Pelagibacter" and *Cyanobacteria*, respectively) show a similar abundance, they present different patterns of recruitment. The most cosmopolitan phages are cyanophages, particularly those that infect the genus *Prochlorococcus*. On the other hand, pelagiphages show a more endemic distribution, especially pelagimyophages, which tend to appear in only a few stations at a time (in this case, as could be expected, in the *Tara* stations in the Mediterranean), while pelagipodophages tend to appear in more stations (Fig. 2). In each of the plots, the recruitment means for each data set were represented as a line, showing that in all three cases (*Alphaproteobacteria*, *Cyanobacteria*, and other phages), the sequences recovered by LR prior to assembly are significantly more abundant than their assembled counterparts (Wilcoxon rank sum test, $P$ value $< 10^{-5}$). Furthermore, this difference in RPKG was accentuated when comparison was made with phages that infect taxa which are typically difficult to assemble, such as those infecting *Alphaproteobacteria* (8). These results suggest that the dereplicated (nonredundant) LR sequences represent an untapped and abundant reservoir of genomic diversity.

Since the phage sequences were obtained from the cell fraction, we were interested to know if they were abundant and could also be recovered in the viral fraction. To that end, we recruited all phage data sets in metagenomes and viromes at different depths obtained at the same location from which the sample was collected (7, 25). When comparing the recruitment values in both types of samples (Fig. 2B), we observed that the vast majority of sequences accumulated significantly more in the viral fraction at the three depths surveyed (Wilcoxon rank sum test, $P$ value $<10^{-16}$, for all three depths). Therefore, we can confirm that the phage genomes recovered from the cellular fraction are representative of the community found in the virion fraction as well and therefore represent a valid method for recovering the viral diversity of a sample.

**New diversity recovered from LR.** Once we discovered that there is a large amount of viral sequences in LR that is not contained in the other data sets (probably lost in the assembly process) and that is abundant in nature, we decided to analyze this diversity. Given that there is no universal marker for analyzing viral diversity, we used a number of

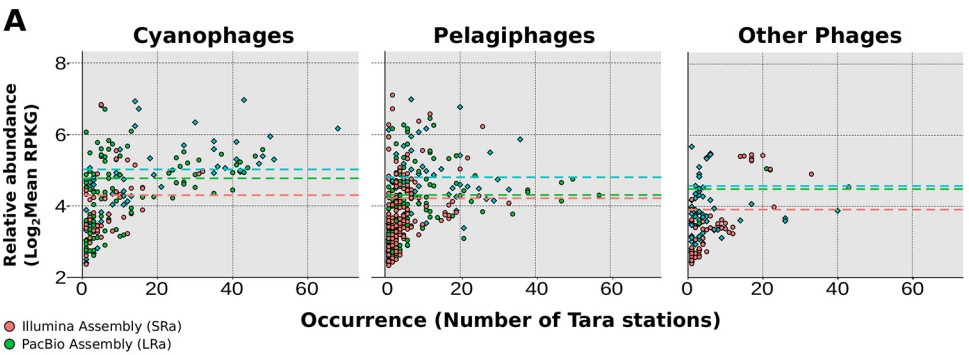

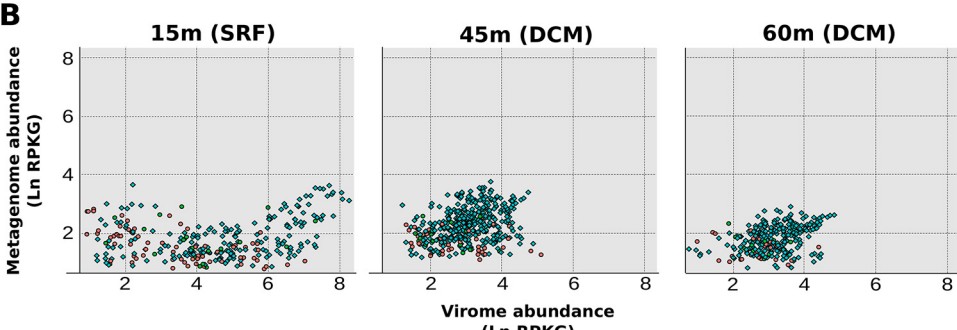

**FIG 2** (A) Relative abundance of viral sequences measured by their recruitment values in metagenomes from *Tara* Oceans expeditions for cyanophages, pelagiphages, and other phages. The *x* axis shows the number of *Tara* stations where the contig accumulated over the coverage thresholds, while the *y* axis shows the combined recruitment value (in RPKG). Circles represent contigs derived from assembly (green for hybrid assembly, orange for Illumina assembly), while blue diamonds represent raw PacBio reads. (B) Relative abundance of viral sequences in viromes (*x* axis) and metagenomes (*y* axis) obtained from the same sample at 15, 45, and 60 m, measured in ln RPKG. Circles represent contigs derived from assembly (green for hybrid assembly, orange for Illumina assembly), while blue diamonds represent raw PacBio CCS15 reads. SRF, surface; DCM, deep chlorophyll maximum.

different phage-specific markers (large terminase subunit [*terL*], replicative DNA helicase [*dnaB*], tail tube protein, major capsid protein, and spanin) as well as several well-characterized auxiliary metabolic genes (AMGs) (thymidylate synthase [*thyX*], phosphoheptose isomerase [*gmhA*], ribonucleoside-diphosphate reductase [*nrdA*], ribonucleotide reductase large subunit, and phosphate starvation-inducible protein [*phoH*]).

We analyzed the diversity of these markers in the same sample for the three data sets by building phylogenetic trees (Fig. 3A; Fig. S2 and S3) and also by comparing the dereplicated sequence distribution with GOV2 (Fig. 3B; Table S2). The phylogenetic trees showed that none of the clades were composed only of LR-unique proteins, so we can conclude that the unique sequences recovered from the LR data set belong not to novel phage taxa but to known clades. Comparing the distribution of unique proteins between our three data sets, the LR data set usually contained more unique sequences by an order of magnitude compared to the assembled data sets (Table S2). Moreover, the percentage of unique variants was always higher in the LR.

After including the GOV2 data set in the comparison, it quickly became apparent that this data set contained most of the unique sequences (ca. 90% of all unique proteins). This was expected, considering the vast size and breadth of sampling of the GOV data set (144 samples); it was therefore surprising that a data set derived from a single sample contains a tenth of the diversity, especially considering that the 10 proteins selected are conserved proteins in phage genomes. Out of this slice of diversity, the vast majority of the unique contigs derive from the unassembled LR data set, as seen in the case of DnaB (149 different proteins versus 26 in the assembled data sets) and RrdA (150 versus 19 in assembled data sets) (Table S2).

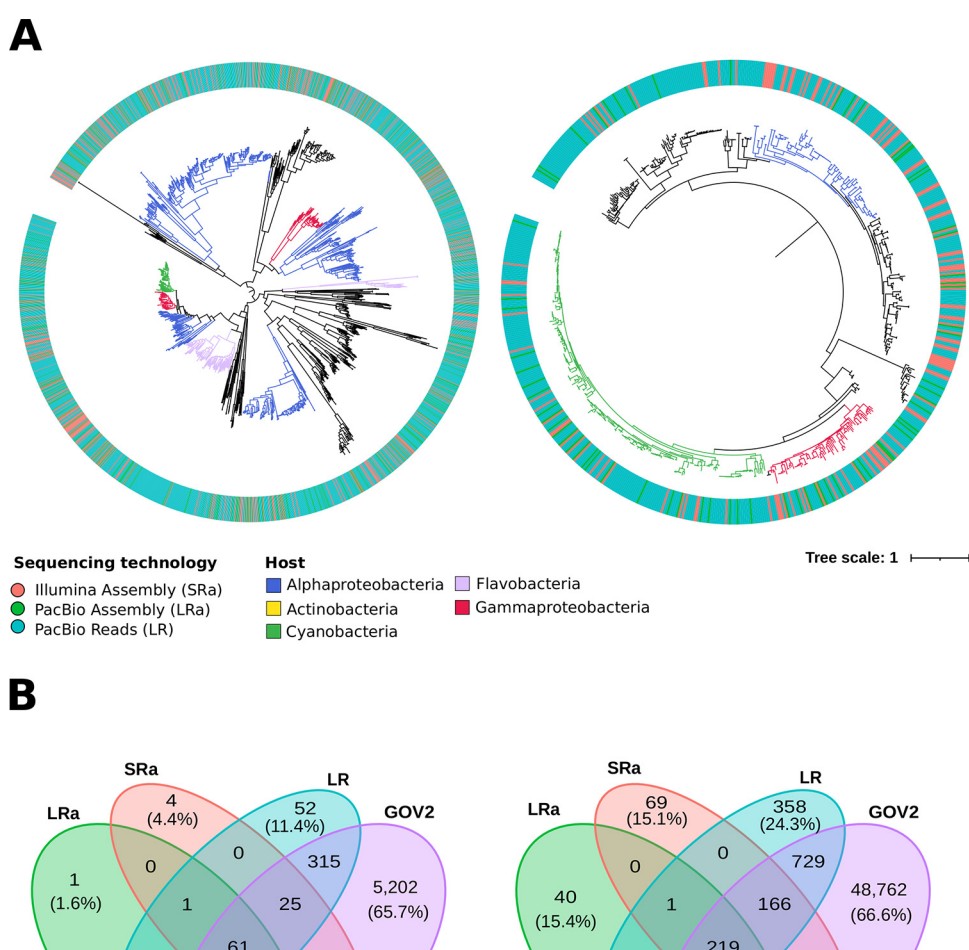

**FIG 3** (A) Phylogenetic trees based on the terminase large subunit (TerL) and thymidylate synthase (PhyX). Branches are colored according to the assigned host, while the color of the outer circle indicates the data set the contig was obtained from (orange for Illumina assembly, green for PacBio assembly, blue for PacBio CCS15 reads). (B) Venn diagrams showing shared and unique sequences among the three data sets and GOV2 for the terminase large subunit (TerL) and thymidylate synthase (PhyX). The number inside each intersection leaf indicates the number of proteins shared by those data sets. In the unique section for each data set, the number in parentheses is the percentage of unique proteins in that data set compared to the total.

It is important to emphasize that the fact that LRs do not reveal novel phage clades does not mean that their novelty is not relevant. An example of this would be the endolysins, a remarkably diverse group of catalytic enzymes that degrade the cell wall of the host so that the phage progeny can escape (26). In recent years, these proteins have awakened increased interest for their potential to be used as antimicrobial agents (27, 28). Culture-free approaches have been applied to great effect in order to broaden the diversity of endolysins. In a previous study (29), 2,628 putative endolysins were retrieved from a collection of 183,298 assembled viral genomes, pooled from a variety of metagenomic data sets. We applied the same pipeline to our samples to evaluate if this novel diversity found by LR would also apply to proteins with more diversity than the usual protein markers.

We recovered 335, 106, and 841 putative endolysins from the SRa, LRa, and LR data sets, respectively, yielding a total of 1,216 new sequences. A phylogenetic tree of the sequences (Fig. S4) reveals that although most of the sequences are distributed among previously described endolysin groups, there were four clades not found in the

previous endolysin environmental collection, which we name C1 to C4. An analysis of their domains revealed them to be glycoside hydrolases from families 24, 104, 23, and again 24, respectively. These are lytic transglycosylases that have the well-known $\alpha + \beta$ lysozyme (30) fold, with differences in activity and specificity thought to be determined by the environment surrounding the active site. Each family includes several well-characterized phage lysozymes. No domains related to cell wall binding were found. Interestingly, the C4 clade contains a signal-arrest-release motif, a mechanism not reported in the original data set (29). This motif first directs the endolysin to the periplasm by first attaching it to the membrane, where it remains inactive until it is released as a soluble active enzyme in the periplasm (31). No other domains related to protein export or cell wall binding were found.

**Functional characterization.** Finally, our last question was if there was any functional category more enriched in the LR data set than in the assemblies. To answer this, we analyzed the protein content at the level of functionality, annotating the proteins against the KEGG (32) and Conserved Domain Database (CDD) (33). Then we compared the number of proteins with each annotation in the LR data set against the proteins found in the assembled data sets. The LR data set was particularly enriched in repeat-containing proteins, such as MORN repeats (37 times higher in LR than in the assembled data sets), pentapeptide repeats (26 times higher), ankyrin repeats (10 times higher), and Kelch repeats (9 times higher). Pentapeptide and Kelch repeats are widespread through bacterial and viral proteins (34, 35), ankyrin repeats have been found in a novel AMG which protects the infected bacteria from eukaryotes (36), and MORN repeats have been found in bacteriophage endolysins (37). The appearance of these proteins was not surprising, as repeats are the main cause of fragmented assemblies (38). A similar argument could be made for the prevalence of integrases (18 times higher), reverse transcriptases (not found in the assembled data sets) and transposases (9 times higher). Although these proteins are widespread in phage genomes (39–41), they present a large amount of microdiversity, which is also difficult for assemblers to solve (12). No groups of proteins were noticeably less abundant in LR than its assembled counterparts. These results suggest that long reads can help recover parts of the viral genome that are difficult to retrieve due to assembly bias.

**Conclusions.** The results obtained here demonstrate that it is possible to recover a representative sample of the viral community fraction of the viral community from the cellular fraction using LR sequencing approaches (e.g., PacBio Sequel II with CCS). This has already been observed with Illumina data sets (15), but the benefits of this approach improve with LR sequencing. The amount of DNA required for a PacBio run is at least an order of magnitude larger than that required for Illumina sequencing, and considering that DNA extraction from the viral fraction is an arduous process, requiring a large amount of sample as well as specialized equipment, studying these recovered viral genomes within the cell size fraction (e.g., $>0.2~\mu m$) may be a good alternative. The benefits of LR sequencing for the study of viral sequences are important even compared with the already-proven advantages for cellular metagenome analysis (12). CCS15 long reads are the equivalent of the average Illumina contig both in terms of length and reliability and therefore allow similarly reliable gene calling and protein identification.

We have also revealed that viral genomic diversity is even greater than previously thought. As the discovery of new endolysins demonstrates, this untapped diversity could aid biotechnological efforts, such as the search for biological agents for medicine and the application of bio-industry to agriculture or food production.

## MATERIALS AND METHODS

**Viral contig recovery and dereplication.** Contigs larger than 1 kb from the three metagenome data sets were described by Haro-Moreno et al. (12). For the LR data set, we decided to analyze the CCS15 data set, consisting of PacBio long reads that have been resequenced at least 15 times. This process results in reads with 99.95% base calling accuracy, which is similar to Illumina error rates and results in accurate gene calling (12). Phage sequence recovery was performed in two steps. Bacteria and archaea viral contigs were recovered using VIBRANT (16) with default parameters. Eukaryotic viruses were recovered via manual curation. Each data set was dereplicated using CD-HIT (42) at 95% identity to remove redundant sequences. Contigs were considered unique based on the definition of "viral population" as

described by Gregory et al. (17); that is, contigs were considered part of the same population if they had hits with at least 95% identity and the sum of distinct alignment lengths resulted in a coverage of at least 70% across the smallest contig using BLASTN (43).

**Genome annotation.** Predicted viral contigs were taxonomically annotated following the method described by Beaulaurier et al. (11). The predicted proteins from each contig were annotated against the NCBI Viral Genomes database (44) (downloaded in September 2021) using LAST (45). Viral contigs were annotated at the order level if they contained one, three, or five or more proteins with top hits to phages that infect the same host genus. The choice of threshold seems to affect only the number of phages classified, not the community composition. The contigs were also functionally annotated following a variation of the method described by Zaragoza-Solas et al. (46). Protein alignments were downloaded from the PHROG (47) and CDD (33) databases and then converted to hidden Markov models (HMMs) using hmmbuild (48). Protein sequences from the three data sets were annotated against the previously built HMMs using hmmscan (48). For each database, we assigned to each gene the best hit with an E value of at least $10^{-5}$ and a query coverage of at least 50%. Proteins were then clustered at 30% identity and 50% query coverage using MMSeqs2 (49), and the annotations for each cluster were manually curated to ensure that the annotations were coherent for all proteins in the cluster. All contigs were searched for the presence of tRNAs using tRNA-scan-SE (50).

**Read recruitment.** Viral contigs from the SRa and LRa data sets and the unique contigs from the SR data set were mapped against the *Tara* Oceans metagenomes using pblat (51), using a cutoff of 95% nucleotide identity over at least 50 nucleotides. Each read was mapped only to the viral contig with the best match. Normalization was performed by calculating RPKG (reads recruited per kilobase of the genome per gigabase of the metagenome) so that recruitment values could be compared across samples.

**Phylogenetic reconstruction of viral marker proteins.** Phylogenetic trees of marker viral proteins were constructed adapting the method described by Benler et al. (52). Marker viral proteins in the SRa, LRa, and LR data sets were detected via hmmsearch (48) against the PHROG (47) database (see "Genome annotation") and merged into a single data set. This data set was then grouped with mmseqs2 (49) into clusters with 50% amino acid identity and a coverage of 70%, which were then aligned using ClustalOmega (53) and compared to each other using hhsearch (48). A distance matrix was calculated by calculating distances following the formula $-\ln[S_{A,B}/\min(S_{A,A}, S_{B,B})]$, where $S_{A,B}$ is the raw score per alignment length. $A$ and $B$ are the different clusters being compared. $S_{A,A}$ and $S_{B,B}$ are the raw alignment score of those clusters aligned to themselves. This matrix was used to build a dendrogram (unweighted pair group method using average linkages [UPGMA]), which acted as a guide to merge clusters using ClustalOmega, resulting in larger protein alignments. The resulting protein alignments were filtered to remove sites with more than 50% gaps and then used to build trees using FastTree (54) (substitution matrix, BLOSUM45; James-Taylor-Thornton model).

**Phylogenetic reconstruction of eukaryotic viruses.** To assess the phylogeny of the contigs categorized as eukaryotic viruses, a concatenation of 5 marker proteins (PolB, SFII, A32, VLTF3, and MCP) was built. ncldv_markersearch (23) was used to identify and align individual marker proteins, and then a Python script was used to build the final concatenation. Contigs were included in the concatenation if they had at least 3 of the 5 marker proteins. A database of 622 NCLDV Metagenome-Assembled Genomes from the previous study (23) and reference NCLDV from RefSeq (55) were added to the concatenation. Finally, a phylogenetic tree was built using IQ-TREE2 (56), using the model VT+F+I+G4 and producing 1,000 ultrafast bootstraps to assess confidence.

**Putative endolysin discovery and analysis.** Putative endolysins in the Mediterranean data sets were extracted following the method described by Fernández-Ruiz et al. (29). The predicted proteins from each contig were compared against a curated database of endolysins using DIAMOND (57). Matches were classified as putative endolysins if the match had >50% identity, covered at least 30% of the query sequence, the alignment was at least 50 aa long and the E value was at least $10^{-3}$. A phylogenetic tree including both the reference data set and new putative sequences was built following the method described above (see "Phylogenetic reconstruction of viral marker proteins"). Protein domains were detected by using hmmsearch (48) against the CDD (33) and dbCAN2 (58) databases, with a match being considered valid if it had 70% HMM coverage and an E value of at least $10^{-5}$. Proteins of the C4 clade were tested for the presence of a signal-arrest-release domain following the method of Oliveira et al. (59).

**Statistical testing.** Wilcoxon rank sum tests were performed using the coin package in R (60). The effect size for Kruskal-Wallis test was calculated using the rstatix package (https://cran.r-project.org/web/packages/rstatix/index.html).

**Data availability.** Metagenomic data sets (Illumina reads, MedWinter-FEB2019-I; PacBio CCS reads, MedWinter-FEB2019-PBCCS15; and PacBio raw reads, MedWinter-FEB2019-PB) are available in the NCBI BioProject database under accession number PRJNA674982.

## SUPPLEMENTAL MATERIAL

Supplemental material is available online only.

**FIG S1**, TIF file, 2.6 MB.

**FIG S2**, TIF file, 2.5 MB.

**FIG S3**, TIF file, 2.2 MB.

**FIG S4**, TIF file, 2 MB.

**TABLE S1**, XLSX file, 0.4 MB.

**TABLE S2**, XLSX file, 0.01 MB.

## ACKNOWLEDGMENTS

This work was supported by the grants VIREVO CGL2016-76273-P [AEI/FEDER, EU] and FLEX3GEN PID2020-118052GB-I00 (cofunded with FEDER funds) from the Spanish Ministerio de Economía, Industria y Competitividad and HIDRAS3 PROMETEU/2019/009 from Generalitat Valenciana. A.Z.-S. was supported by a Ph.D. fellowship from the Spanish Ministerio de Economía y Competitividad (BES-2017-079993).

M.L.-P. and F.R.-V. conceived the study. A.Z.-S. analyzed the data. J.M.H.-M. and F.R.-V. contributed to writing the manuscript. All authors revised the manuscript and approved the final version.

We declare that we have no competing interests.

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
