## [Reviewer comments · mSystems]

Long-read metagenomic improves the recovery of viral diversity from complex natural marine samples

Asier Zaragoza-Solas, José Haro-Moreno, Francisco Rodríguez-Valera, and Mario López-Pérez

Corresponding Author(s): Mario López-Pérez, Universidad Miguel Hernandez

Review Timeline:

Submission Date:	February 24, 2022
Editorial Decision:	April 12, 2022
Revision Received:	April 27, 2022
Editorial Decision:	May 6, 2022
Revision Received:	May 9, 2022
Accepted:	May 11, 2022

Editor: Julie Huber

Reviewer(s): The reviewers have opted to remain anonymous.

Transaction Report:

DOI: <https://doi.org/10.1128/msystems.00192-22>

April 12, 2022

Dr. Mario López-Pérez
Universidad Miguel Hernandez
Producción Vegetal y Microbiología
Campus de San Juan
Apartado 18
Alicante, Alicante 03550
Spain

Re: mSystems00192-22 (Long-read metagenomic improves the recovery of viral diversity from complex natural marine samples)

Dear Dr. Mario López-Pérez:

Thank you for submitting your manuscript to mSystems. We have completed our review and I am pleased to inform you that, in principle, we expect to accept it for publication in mSystems. However, acceptance will not be final until you have adequately addressed the reviewers comments. Both reviewers requested more specific statistical and quantitative analysis, in particular, and more details of analyses. I also note you need to provide a paragraph on data availability, per mSystems policies.

Preparing Revision Guidelines

Sincerely,

Julie Huber

Editor, mSystems

Journals Department
American Society for Microbiology

Reviewer comments:

Reviewer #1 (Comments for the Author):

General comments:

Third generation long-read sequencing has the potential to overcome assembly biases resulting from second-gen short-read approaches. Here, the authors report a hybrid approach incorporating both short- and long-read sequencing to recover viral genomes from a cellular fraction metagenomic sample from offshore Mediterranean waters. The authors contrast their results with short-read-assembly only approaches and found that the hybrid approach recovered genomic completion and long-read (LR)-only approach increased recovery of sequence diversity. Overall, I enjoyed reading this paper and found it timely and relevant to recovering microbial diversity from natural habitats, clear, and well-written.

My major concerns center around the reported novelty/sequence diversity of the dataset that is central to the purpose of the paper of comparing and contrasting different sequencing/assembly methods. The methods reported do not convince me that key reported numbers are reliable or relevant to the questions at hand - a revision incorporating these comments and/or clarification on the methods would bring me on board. I looked for more information in the Supplementary, but no additional details on methods were included.

Major considerations:

1. Reported number of novel viruses recovered by long read (LR) only approach vs. long-read assemblies (LRa) and short-read assemblies (SRa)

Lines 175-180: the authors state that the LR-recovered viruses had much smaller length (5kb) than the SRa and LRa datasets (eg. LRa: 32kb). These differences are problematic when applying the same de-replication cut-off to identify unique sequences. Considering that the avg viral genome size in the surface ocean is 40kb, a 40kb genome can result in 8 "unique" 5kb viruses. Furthermore, 95% over 70% alignment cut-off used for de-replication means that a 40kb genome can result in $40/5/0.7 = 11.4$ "unique" 5kb viruses.

I'd find the above comparison (how many novel viruses using different methods) more convincing using a marker gene-based approach eg. using the gene sequences precluding the terminase protein tree analysis already done here in Fig. 3. This analysis gets around inflated numbers due to sequence length differences amongst different methods. One can also think of some corrections/normalizations based on length

Line 158 and 392: does the 95% identity over 70% alignment cutoff to de-replicate viral sequences stipulate that the alignment must be in one contiguous piece, or is the 70% summed across the entire shorter contig? If this alignment cutoff is one piece, then 70% of the contig inflates the number of "unique" viral sequences as it does not account for circular permutations. Circular permutations in the start/end of viral genomes mean that viruses in the same population can share anywhere from 50-100% of their sequences in a contiguous alignment.

Line 390-391: "if they had no hits to other contigs": Does this mean that multiple sequences that shared >95% ID across >70% with two other sequences will not be clustered together? I don't recall that this is how cd-hit-est works, but the wording here suggests otherwise.

2. Recovery relative to the GOV dataset

Lines 165-173: I find this section lacking in reference databases if the goal is "assessing if this new found diversity had been captured by previous studies". Although GOV is the largest database to date, it is limited to the cellular metagenomes that contain a lower relative abundance of viruses relative to viromes. Furthermore, this dataset is limited to only the near-surface ocean. To comprehensively assess "if this new found diversity had been captured by previous studies", I'd like to see a comparison with viral database representation including both cellular and viral metagenomes capturing previously reported viral diversity from diverse planktonic habitats: Earth Viromes, ALOHA viral database, UVMed, UVDeep, Pacific Ocean Viromes, etc. The same workflow could be applied to these datasets to more comprehensively subtract the previously reported viral diversity from this dataset - then I'd find the numbers here convincing with regards to revealing the # of viral sequences that were distinct from anything sequenced before.

Line 265-268 and read-recruitment analyses: are these abundances based on all hits or best hits for read-mapping? Best hits

(eg. one read can only map to one viral sequence) will yield a more accurate metric of relative abundances, but I cannot find this specified in the methods (lines 412-417).

Moderate considerations:

Lines 47-51: I'm not understanding why cellular metagenomic sequencing is "more efficient than the classical virome sequencing" when viral sequences account for a higher proportion of the "virome" size fraction compared to cellular sized fractions (see Fig 1 <https://www.nature.com/articles/s41396-020-0604-8>).

Line 79-81: I find this statement inaccurate considering the at least 2 (possibly +) marine papers so far cited here, and likely more examples from other environments: "The problem is that the high error rate derived from these technologies has not made them suitable for application in metagenomics so far."

Lines 153-154: Interesting - I'd like to hear thoughts on why assemblies might be biased towards low-GC contigs

Lines 190-191: are these numbers de-replicated/non-redundant? "number of proteins recovered from the LR dataset are an order of magnitude larger than in the assembled datasets"

Line 234-249: Cool work and neat examples of the type of novel diversity LR/hybrid approaches could recover

364: I'd like to see this "considerable fraction of the viral community from the cellular fraction" specified

Minor comments:

Line 34: This sentence is unclear to me for contrasting the reported numbers - does "comparison" here means overlap between the two datasets?

Line 50: I'm not sure what "efficient" here means - perhaps use more specific wording here?

Line 94: typo add "of" between 'study viruses'

Line 159: typo "Lra"

Line 200 & 395 & possibly others: typo in author name

Lines 269-270: "difficult to assemble phage taxa, such as those infecting Alphaproteobacteria": where is this based from?

Line 304: "it" is unclear here

Reviewer #2 (Comments for the Author):

The authors describe using a combination of short and long read sequencing technologies on a sample of seawater that was collected on a 0.2 um filter, therefore representing possible viruses infected or associated with their hosts with the sample. The use of LR technology has been performed by others as the authors reference (although there are more outside of marine samples), yet the authors set out to determine the differences between these technologies and their potential influence on data recovered. I found the functional characterization the most interesting as it provided empirical evidence for the differences (largely due to the long-known issue with assembly-based approaches breaking or being unable to assemble sequences with repetitive contiguous nucleotides).

As the authors were reporting on the use and comparison of these technologies, the manuscript lacked sufficient statistical rigor (I have provided a few examples below, though it is not exhaustive). Due to this, it often felt the analyses were observational or qualitative rather than quantitative.

Other comments:

Line 32: CCS - write out its meaning

Importance: The authors use an established technique to examine recovered sequences for viral homology, rather than 'developed an approach'.

Line 143: The authors say the recovery of viral sequences was 'significant' at 5% and 2.5%, the authors need to provide the

statistical evaluation used to determine significance.

Line 155: 'light skew' can the authors provide the skew to understand the basis of 'skew'.

Lines 173 - 176: How did the authors determine relatedness (was it 90% over 70% of sequence)? Also, the likely reason more were identified in the LR is due to the number of sequences recovered from LR versus the other datasets. The authors should also provide a normalized value.

Also, in Lines 230 - 243: a normalized value for comparisons would be useful.

Line 269: Does 'podophages' refer to pelagiphages that are in the family Podoviridae?

Line 283: "Significantly more", It is unclear what the significance is referring to and what the authors are describing - they recovered more in what way?

Line 289: Is it diversity lost or total number of contigs that were viral, which I suppose could be a richness factor, but it is unclear what the authors used as a diversity metric.

Line 393: "Bacteria and archaea contigs" or bacteria and archaea virus contigs?

General comments: There are spelling and issues with punctuation (0.2 μm (Line 26) and e.g., (Line 42), respectively) and possible spaces before periods that are not needed, though I cannot tell if this is an issue or a consequence of using 'justify' for the document alignment.

Line 161: 'Lra' should be LRa.

Line 195: 'CCS15' do the authors mean CCS?

Table 1: The sequencing depth is vastly different, yet not accounted for in the analyses presented.

Reviewer comments:

Reviewer #1 (Comments for the Author):

General comments:

Third generation long-read sequencing has the potential to overcome assembly biases resulting from second-gen short-read approaches. Here, the authors report a hybrid approach incorporating both short- and long-read sequencing to recover viral genomes from a cellular fraction metagenomic sample from offshore Mediterranean waters. The authors contrast their results with short-read-assembly only approaches and found that the hybrid approach recovered genomic completion and long-read (LR)-only approach increased recovery of sequence diversity. Overall, I enjoyed reading this paper and found it timely and relevant to recovering microbial diversity from natural habitats, clear, and well-written.

My major concerns center around the reported novelty/sequence diversity of the dataset that is central to the purpose of the paper of comparing and contrasting different sequencing/assembly methods. The methods reported do not convince me that key reported numbers are reliable or relevant to the questions at hand - a revision incorporating these comments and/or clarification on the methods would bring me on board. I looked for more information in the Supplementary, but no additional details on methods were included.

Authors: We appreciate the positive comments. We have addressed, to the best of our knowledge, every point raised by the reviewers and included their suggestions in the manuscript. Point by point responses can be found below.

Major considerations:

1. Reported number of novel viruses recovered by long read (LR) only approach vs. long-read assemblies (LRa) and short-read assemblies (SRa)

Lines 175-180: the authors state that the LR-recovered viruses had much smaller length (5kb) than the SRa and LRa datasets (eg. LRa: 32kb). These differences are problematic when applying the same de-replication cut-off to identify unique sequences. Considering that the avg viral genome size in the surface ocean is 40kb, a 40kb genome can result in 8 "unique" 5kb viruses. Furthermore, 95% over 70% alignment cut-off used for de-replication means that a 40kb genome can result in $40/5/0.7 = 11.4$ "unique" 5kb viruses.

I'd find the above comparison (how many novel viruses using different methods) more convincing using a marker gene-based approach eg. using the gene sequences precluding the terminase protein tree analysis already done here in Fig. 3. This analysis gets around inflated numbers due to sequence length differences amongst different methods. One can also think of some corrections/normalizations based on length

Authors: We agree with the concerns raised by the reviewer; however, they are an unfortunate consequence of being limited to work with phage fragments instead of complete genomes when analysing viromes. We have taken these limitations into account, and for this reason, the methods used in this paper have been used to determine phage diversity in a

number of high-profile papers analysing virome datasets (PMID: 25999515; PMID: 27654921; PMID: 30556814). The use of a marker gene is not without drawbacks, namely that even highly conserved phage genes cannot be reliably identified in all phage genomes, thus also underrepresenting phage diversity.

However, as the main discovery reported in the paper is the diversity found in long read datasets, we understand that further evidence that circumvents this issue is required. We have added the unique terminase counts (based on protein clustering, 95% identity) found in all three datasets to Table 1 and updated the text to provide more weight to our argument. We have added the sentence: "This diversity gap was also present when comparing a marker gene such as the terminase large subunit (terL), with the LR dataset containing 393 unique terminase genes compared to 30 and 2 in the SR and LRa datasets (protein clustering at 95% sequence identity), respectively."

Line 158 and 392: does the 95% identity over 70% alignment cutoff to de-replicate viral sequences stipulate that the alignment must be in one contiguous piece, or is the 70% summed across the entire shorter contig? If this alignment cutoff is one piece, then 70% of the contig inflates the number of "unique" viral sequences as it does not account for circular permutations. Circular permutations in the start/end of viral genomes mean that viruses in the same population can share anywhere from 50-100% of their sequences in a contiguous alignment.

Authors: The alignment is summed across the entire shorter contig, taking care of possible repeat regions. The relevant materials and methods section has been expanded to reflect this. The sentence now reads: "contigs were considered part of the same population if they had hits with at least 95% identity and the sum of distinct alignment lengths resulted in a coverage of at least 70% across the smallest contig using BLASTN".

Line 390-391: "if they had no hits to other contigs": Does this mean that multiple sequences that shared >95% ID across >70% with two other sequences will not be clustered together? I don't recall that this is how cd-hit-est works, but the wording here suggests otherwise.

Authors: The reviewer is right, multiple sequences that share similarity over the threshold would be clustered into the same viral population. The relevant section has been updated to avoid this misunderstanding

2. Recovery relative to the GOV dataset

Lines 165-173: I find this section lacking in reference databases if the goal is "assessing if this new found diversity had been captured by previous studies". Although GOV is the largest database to date, it is limited to the cellular metagenomes that contain a lower relative abundance of viruses relative to viromes. Furthermore, this dataset is limited to only the near-surface ocean. To comprehensively assess "if this new found diversity had been captured by previous studies", I'd like to see a comparison with viral database representation including both cellular and viral metagenomes capturing previously reported viral diversity from diverse planktonic habitats: Earth Viromes, ALOHA viral database, UVMed, UVDeep, Pacific Ocean Viromes, etc. The same workflow could be applied to these datasets to more comprehensively subtract the previously reported viral diversity from this dataset - then I'd

find the numbers here convincing with regards to revealing the # of viral sequences that were distinct from anything sequenced before.

Authors: We do not understand the reviewer's comment. The Global Ocean Viromes 2 (GOV2) paper (PMID: 31031001) clearly states that it is built upon the previous GOV database, which was based on TARA and Malaspina viral fraction sample datasets, and that the new 41 arctic samples are also viromes. Therefore, the cellular fraction is not contained in this dataset.

Considering that the datasets analysed in this study are derived from marine surface water samples, we believe that GOV2 is the best dataset to assess their diversity, as it contains the vast majority of samples that cover phage diversity from all around the globe from environments similar to our sample, and are based on shotgun metagenomics with an Illumina platform (also used on this work). The datasets proposed by the reviewer are either based on a different approach (UVMed and UVDeep are fosmid collections, not shotgun sequencing, Pacific Ocean Viromes uses 454 instead of Illumina), contain samples not relevant to our sample (Earth Virome contains samples from a variety of environments, not only marine) or are datasets that are limited in scope (the ALOHA viral database is exclusively composed of samples from Station ALOHA).

Line 265-268 and read-recruitment analyses: are these abundances based on all hits or best hits for read-mapping? Best hits (eg. one read can only map to one viral sequence) will yield a more accurate metric of relative abundances, but I cannot find this specified in the methods (lines 412-417).

Authors: They are abundance values based on best hits. The manuscript has been updated to include more details. In the materials and methods section, we have included the sentence: "Each read was mapped only to the viral contig with the best match."

Moderate considerations:

Lines 47-51: I'm not understanding why cellular metagenomic sequencing is "more efficient than the classical virome sequencing" when viral sequences account for a higher proportion of the "virome" size fraction compared to cellular sized fractions (see Fig 1 <https://www.nature.com/articles/s41396-020-0604-8>).

Authors: Our intention was to state the efficiency of long-read sequencing over short-read sequencing for the analysis of phage sequences from metagenomes, not that phage recovery is more efficient in the cellular fraction than in the viral fraction. We have updated the sentence "This approach results in the recovery of a representative sample of the viral population and it has proven to better perform (more phage contigs, larger average contig size) than Illumina sequencing applied to the same sample." We hope it is clearer now.

Line 79-81: I find this statement inaccurate considering the at least 2 (possibly +) marine papers so far cited here, and likely more examples from other environments: "The problem is that the high error rate derived from these technologies has not made them suitable for application in metagenomics so far."

Authors: The reviewer is right. It would be more correct to state that its application in metagenomics has been delayed until error-correcting methods were available. The text has been updated to correct this "Unfortunately, the high error rate derived from these technologies has delayed its application in metagenomics, and requires the use of either a short-read dataset (10) or a high-coverage dataset (11) to correct these sequencing errors.". We hope it is clearer now.

Lines 153-154: Interesting - I'd like to hear thoughts on why assemblies might be biased towards low-GC contigs

Authors: Our hypothesis is that this bias arises from the fact that assemblies usually only recover the core genome. In this sample (marine surface water) SAR11 is the most abundant organism, with an average GC content of 34%. Long reads recover more of the flexible genome, which can present GC fluctuations compared to the core and would thus explain this variation from 34 to 38%, but we cannot prove it.

Lines 190-191: are these numbers de-replicated/non-redundant? "number of proteins recovered from the LR dataset are an order of magnitude larger than in the assembled datasets"

Authors: The protein sequences have been dereplicated (CD-HIT, 95% identity threshold). Table 1 has been updated to include this information.

Line 234-249: Cool work and neat examples of the type of novel diversity LR/hybrid approaches could recover

Authors: Thank you very much for this positive comment

364: I'd like to see this "considerable fraction of the viral community from the cellular fraction" specified

Authors: This sentence refers to the results described in "Relative Abundance in marine samples", in which phages obtained from the cellular fraction also recruit in the viral fraction. The text has been updated to better reflect this and now it is "The results obtained demonstrate that it is possible to recover a representative sample of the viral community fraction of the viral community from the cellular fraction using LR sequencing approaches"

Minor comments:

Line 34: This sentence is unclear to me for contrasting the reported numbers - does "comparison" here mean overlap between the two datasets?

Authors: It does mean overlap. The sentence has been updated for better clarity."...and ca. 26,000 had no homologues to the large dataset of the Global Ocean Virome 2 (GOV2)". We hope it is clearer now.

Line 50: I'm not sure what "efficient" here means - perhaps use more specific wording here?

Authors: We meant the quality of the recovered phage dataset - more and larger phage contigs. We have updated the text to be more specific, as requested. The sentence is now "This approach results in the recovery of a representative sample of the viral population and it has proven to perform better (more phage contigs, larger average contig size) than Illumina sequencing applied to the same sample." We hope it is clearer now.

Line 94: typo add "of" between 'study viruses'

Authors: Fixed.

Line 159: typo "Lra"

Authors: Fixed.

Line 200 & 395 & possibly others: typo in author name

Authors: Fixed.

Lines 269-270: "difficult to assemble phage taxa, such as those infecting Alphaproteobacteria": where is this based from?

Authors: There is a mistake in this sentence: the differences in RPKG were accentuated in phages that infect taxa which are typically difficult to assemble, not that the phages themselves were difficult to assemble. The sentence has been rewritten to reflect this and a reference has been added. "Furthermore, this difference in RPKG was accentuated when comparing phages that infect taxa which are typically difficult to assemble, such as those infecting Alphaproteobacteria"

Line 304: "it" is unclear here

Authors: "it" means "GOV2 dataset". We have changed this in the sentence. We hope it is clearer now.

Reviewer #2 (Comments for the Author):

The authors describe using a combination of short and long read sequencing technologies on a sample of seawater that was collected on a 0.2 um filter, therefore representing possible viruses infected or associated with their hosts with the sample. The use of LR technology has been performed by others as the authors reference (although there are more outside of marine samples), yet the authors set out to determine the differences between these technologies and their potential influence on data recovered. I found the functional characterization the most interesting as it provided empirical evidence for the differences (largely due to the long-known issue with assembly-based approaches breaking or being unable to assemble sequences with repetitive contiguous nucleotides.

As the authors were reporting on the use and comparison of these technologies, the manuscript lacked sufficient statistical rigor (I have provided a few examples below, though it

is not exhaustive). Due to this, it often felt the analyses were observational or qualitative rather than quantitative.

Authors: We are very appreciative of the positive comments and the thoughtful suggestions. We believe they have significantly strengthened the manuscript. Please find below our point by point answers to them and how we have addressed them in the main text.

Other comments:

Line 32: CCS - write out its meaning

Authors: Fixed.

Importance: The authors use an established technique to examine recovered sequences for viral homology, rather than 'developed an approach'.

Authors: Fixed. We have removed "developed an approach" from the sentence.

Line 143: The authors say the recovery of viral sequences was 'significant' at 5% and 2.5%, the authors need to provide the statistical evaluation used to determine significance.

Authors: We used "significant" as a way to emphasise the large amount of viral sequences found in the cellular fraction, and we never intended to imply that the differences in viral contig amounts between LR and SRa datasets. However, the reviewer is correct that there is no statistical evaluation, and we understand that "significant" could be misinterpreted as "statistically significant". We have updated the text to reflect this. The sentence now is "Viral sequences turned out to be quite numerous in both datasets, with 5% of the total sequences from the SRa and LRa and 2.5% of the LR dataset being classified as viral contigs." It would be interesting to confirm if these differences are indeed significant or not, but the number of LR metagenomic datasets available in public repositories is still too low to have a representative sample size.

Line 155: 'light skew' can the authors provide the skew to understand the basis of 'skew'.

Authors: We have added numerical values to the text. The sentence is now "The SRa dataset, containing only assembled Illumina reads, presented an average GC content of 35.45%; while the hybrid assembly of Illumina and PacBio reads (LRa dataset) showed an average GC content of 36.9%. The LR dataset, containing only PacBio reads, presented a GC content of 38.13%." We hope it is clearer now.

Lines 173 - 176: How did the authors determine relatedness (was it 90% over 70% of sequence)? Also, the likely reason more were identified in the LR is due to the number of sequences recovered from LR versus the other datasets. The authors should also provide a normalized value.

Authors: Relatedness was determined by sequence similarity (95% over 70% of the shorter sequence). The alignment is summed across the entire shorter contig, taking care of possible repeat regions. The text has been updated to provide a more detailed explanation. The sentence now reads: "contigs were considered part of the same population if they had

hits with at least 95% identity and the sum of distinct alignment lengths resulted in a coverage of at least 70% across the smallest contig using BLASTN". In the case of sequence counts, the sequencing effort is not comparable for both SR and LR sequencing and they use the same sample (See PMID: 34512586). The LR dataset is much larger because it consists of corrected, dereplicated raw reads directly while the SRa and LRa use contigs obtained from assembly. The values provided could be normalised by raw nucleotide sequenced counts (23.4Gb for SR, 7.4 for LR CCS15), but we decided against it because we are comparing different datasets produced by vastly different technologies that have undergone different processes, including a dataset that used both types of reads (The hybrid assembly LRa).

Also, in Lines 230 - 243: a normalized value for comparisons would be useful.

Authors: See the above response in regards to normalizations between LR/SR datasets. We have removed the word "significant" from the text to avoid confusion.

Line 269: Does 'podophages' refer to pelagiphages that are in the family Podoviridae?

Authors: It does. We have updated the text to better reflect this. we have changed the names to "pelagimyophages" and "pelagipodophages" to avoid confusion.

Line 283: "Significantly more", It is unclear what the significance is referring to and what the authors are describing - they recovered more in what way?

Authors: We refer to the fact that the phage contigs recovered are more abundant in the viral fraction than in the cellular fraction, determined by recruitment values. In this case we have added statistical data to denote significance using Wilcoxon rank sum test. We have also included the description in the materials and methods section.

Line 289: Is it diversity lost or total number of contigs that were viral, which I suppose could be a richness factor, but it is unclear what the authors used as a diversity metric.

Authors: We have changed the sentence and now it is "Once we discovered that there is a larger amount of viral sequences in LR not contained in the other datasets, probably lost in the assembly process, and that is abundant in nature, we decided to analyse this diversity. Given that there is no universal marker for analyzing viral diversity, we use a number of different phage-specific markers". We hope it is clearer now.

Line 393: "Bacteria and archaea contigs" or bacteria and archaea virus contigs?

Authors: It means "Bacterial and archaea viral contigs". Fixed.

General comments: There are spelling and issues with punctuation (0.2 µm (Line 26) and e.g., (Line 42), respectively) and possible spaces before periods that are not needed, though I cannot tell if this is an issue or a consequence of using 'justify' for the document alignment.

Authors: We have fixed the punctuation issues. The spaces seem to be a formatting issue.

Line 161: 'Lra' should be LRa.

Authors: Fixed.

Line 195: 'CCS15' do the authors mean CCS?

Authors: CCS15 means the subset of PacBio CCS reads that are derived from at least 15 subreads, which results in an estimated error rate similar to those in Illumina sequencing. An explanation has been added to the paper to avoid confusion. “In order to evaluate the possible biases introduced by the assembly process, we also analysed the PacBio CCS15 reads (LR), that is, PacBio consensus reads created by comparing at least 15 subreads, before assembly.”

Table 1: The sequencing depth is vastly different, yet not accounted for in the analyses presented.

Authors: As we have explained above, the sequencing depth is not vastly different (See PMID: 34512586). In fact, the number of raw reads is much higher in the Illumina dataset (23.4Gb for SR, 7.4 for LR CCS15). The differences in sequence count is due to the fact that the LR dataset uses corrected, dereplicated raw reads directly while the SRa and LRa use contigs obtained from assembly. That is the great advantage of LRs that it does not require the assembly step that seems to introduce many biases.

May 6, 2022

Dr. Mario López-Pérez
Universidad Miguel Hernandez
Producción Vegetal y Microbiología
Campus de San Juan
Apartado 18
Alicante, Alicante 03550
Spain

Re: mSystems00192-22R1 (Long-read metagenomic improves the recovery of viral diversity from complex natural marine samples)

Dear Dr. Mario López-Pérez:

Thank you for submitting your manuscript to mSystems. We have completed our review and I am pleased to inform you that, in principle, we expect to accept it for publication in mSystems. However, acceptance will not be final until you have adequately addressed the second reviewer comments. This reviewer believes, and I tend to agree, that many of the statements are qualitative assessments like 'majority', 'more', 'increased' etc - and without statistical rigor routinely. Please address that in your response.

Preparing Revision Guidelines

Sincerely,

Julie Huber

Editor, mSystems

Journals Department
Reviewer comments:

Reviewer #1 (Comments for the Author):

Thank you for addressing my considerations

Reviewer #2 (Comments for the Author):

I appreciate the authors efforts in the revised manuscript. I still feel that while the manuscript has unique and valuable data and insights, additional statistical support or validation is warranted. I offer a few points below, if the editor/authors find them useful.

Concerning normalization, thank you for including PMID: 34512586 in your response. However, please see figure 4, the 'Values are normalized by the assembly size and sequencing effort (blue and red bars, respectively).' This is what way I am referring in terms of normalizing the data in appropriate way(s) to be compared (e.g., MB of MAGs (or contigs used in your analysis)/MB Assembled). For instance, reporting on the whether a majority of contigs were in one dataset or the other should be normalized first (lines 151-154 of original manuscript).

Thank you for adding the skew, here the authors could report, based on simple statistical analysis if the difference is significant, yet I do not see that they do. If it is in fact statistically different that is an important point. Even if it is not, this information is important from a computational standpoint.

For the CCS15, I should have been more clear, however was it stated in the methods or in results why the CCS15 were chosen (I did not find this information)? Perhaps it based on the previous report (reference 10)? Or, was it validated here (as they did in the previous reference). Further, as the authors conclude (line 382) the CCS15 were the same as Illumina contig reliability. I'm not clear on what is meant by reliability.

Reviewer #1 (Comments for the Author):

Thank you for addressing my considerations.

Reviewer #2 (Comments for the Author):

I appreciate the authors efforts in the revised manuscript. I still feel that while the manuscript has unique and valuable data and insights, additional statistical support or validation is warranted. I offer a few points below, if the editor/authors find them useful.

Concerning normalization, thank you for including PMID: 34512586 in your response. However, please see figure 4, the 'Values are normalized by the assembly size and sequencing effort (blue and red bars, respectively).' This is what way I am referring in terms of normalizing the data in appropriate way(s) to be compared (e.g., MB of MAGs (or contigs used in your analysis)/MB Assembled). For instance, reporting on the whether a majority of contigs were in one dataset or the other should be normalized first (lines 151-154 of original manuscript).

Authors: The main issue here is that the datasets used in this study are different enough that there is not a single variable appropriate to normalise. In Figure 4 of PMID: 34512586 the three datasets are assembled contigs, while in this paper we are comparing two assemblies and one raw read dataset. We have added a normalised value of unique sequences to table 1. We don't consider these values to be entirely correct, as the size of the assembly is not strongly correlated to the size of the read dataset, but it is the best option available.

Thank you for adding the skew, here the authors could report, based on simple statistical analysis if the difference is significant, yet I do not see that they do. If it is in fact statistically different that is an important point. Even if it is not, this information is important from a computational standpoint.

Authors: We have added statistical significance and effect size values to the text. The text now reads: "The GC content showed a light (effect size = 0.022) but significant (Kruskal-Wallis test, p-value < 10^{-15}) skew towards high GC values when PacBio CCS reads were added to the datasets (Table 1)."

For the CCS15, I should have been more clear, however was it stated in the methods or in results why the CCS15 were chosen (I did not find this information)? Perhaps it based on the previous report (reference 10)? Or, was it validated here (as they did in the previous reference). Further, as the authors conclude (line 382) the CCS15 were the same as Illumina contig reliability. I'm not clear on what is meant by reliability.

Authors: The reasoning for choosing CCS15 is based on the findings of the previous report. We chose CCS15 because this amount of correction passes results in PacBio reads with the same error rate as Illumina reads (99.95% base call accuracy), which in turn results in accurate gene calling directly from the long reads. The text has been updated to include this information: "For the LR dataset, we decided to analyse the CCS15 dataset, consisting of PacBio long reads that have been resequenced at least 15 times. This process results in

reads with 99.95% base call accuracy, which is similar to Illumina error rates and results in accurate gene calling (12).”

May 11, 2022

Dr. Mario López-Pérez
Universidad Miguel Hernandez
Producción Vegetal y Microbiología
Campus de San Juan
Apartado 18
Alicante, Alicante 03550
Spain

Re: mSystems00192-22R2 (Long-read metagenomic improves the recovery of viral diversity from complex natural marine samples)

Dear Dr. Mario López-Pérez:

Your manuscript has been accepted, and I am forwarding it to the ASM Journals Department for publication. For your reference, ASM Journals' address is given below. Before it can be scheduled for publication, your manuscript will be checked by the mSystems production staff to make sure that all elements meet the technical requirements for publication. They will contact you if anything needs to be revised before copyediting and production can begin. Otherwise, you will be notified when your proofs are ready to be viewed.

Publication Fees:

We recognize that the video files can become quite large, and so to avoid quality loss ASM suggests sending the video file via <https://www.wetransfer.com/>. When you have a final version of the video and the still ready to share, please send it to mSystems staff at mSystems@asmusa.org.

For mSystems research articles, if you would like to submit an image for consideration as the Featured Image for an issue, please contact mSystems staff at mSystems@asmusa.org.

Sincerely,

Julie Huber
Editor, mSystems

Journals Department
Figure S2: Accept
Figure S1: Accept
Supplemental Material: Accept
Supplemental Material: Accept
Figure S4: Accept
Figure S3: Accept